# Improved Antioxidant, Anti-inflammatory, and Anti-adipogenic Properties of Hydroponic Ginseng Fermented by *Leuconostoc mesenteroides* KCCM 12010P

**DOI:** 10.3390/molecules24183359

**Published:** 2019-09-16

**Authors:** Ji Eun Hwang, Kee-Tae Kim, Hyun-Dong Paik

**Affiliations:** 1Department of Food Science and Biotechnology of Animal Resources, Konkuk University, Seoul 05029, Korea; vivavima@naver.com; 2Food Biotechnology Research Institute, Konkuk University, Seoul 05029, Korea; richard44@hanmail.net

**Keywords:** anti-adipogenic effect, anti-inflammatory effect, antioxidative effect, hydroponic ginseng, fermentation, *Panax ginseng*

## Abstract

Hydroponic ginseng (HPG) has been known to have various bio-functionalities, including an antioxidant effect. Recently, fermentation by lactic acid bacteria has been studied to enhance bio-functional activities in plants by biologically converting their chemical compounds. HPG roots and shoots were fermented with *Leuconostoc mesenteroides* KCCM 12010P isolated from kimchi. The total phenolic compounds, antioxidant, anti-inflammatory, and anti-adipogenic effects of these fermented samples were evaluated in comparison with non-fermented samples (control). During 24 h fermentation of HPG roots and shoots, the viable number of cells increased to 7.50 Log colony forming unit (CFU)/mL. Total phenolic and flavonoid contents of the fermented HPG roots increased by 107.19% and 645.59%, respectively, compared to non-fermented HPG roots. The antioxidant activity of fermented HPG, as assessed by 2,2′-azino-bis-3-ethylbenzothiazoline-6-sulfonic acid (ABTS), β-carotene-linoleic, and ferric reducing antioxidant power (FRAP) assay, was also significantly enhanced. In an anti-inflammatory effect of lipopolysaccharide (LPS)-stimulated RAW 264.7 cells, the nitric oxide content and the expression of inducible nitric oxide synthase (iNOS), tumor necrosis factor-α (TNF-α), interleukin-1β (IL-1β), and interleukin-6 (IL-6) decreased when treated with fermented samples. Simultaneously, lipid accumulation in 3T3-L1 adipocyte was reduced when treated with fermented HPG. Fermentation by *L. mesenteroides* showed improved antioxidant, anti-inflammatory and anti-adipogenic HPG effects. These results show that fermented HPG has potential for applications in the functional food industry.

## 1. Introduction

Korean *Panax ginseng* Meyer has been used as a functional food source. The bio-functionality of ginseng, including antioxidation and immune enforcing properties, is derived from various phytochemicals in ginseng, such as numerous kinds of ginsenosides [1], glycosylic acid, and polyphenolic compounds. It takes several years to cultivate ginseng in soil, but there are several obstacles including diseases as well as the accumulation of residual heavy metals and pesticides during cultivation. Recently, studies on hydroponic ginseng (HPG), which is cultivated in mineral nutrient solutions, has performed increasingly to overcome these obstacles. Additionally, the management cost of a hydroponic cultivation technique is much less time-consuming and more economical than that of the conventional soil-cultivation technique. Very few studies, comparing the functional effects of HPG to those of ginseng cultivated in soil, exist but some researchers reported anti-oxidative and anti-inflammatory properties of hydroponic ginseng and soil-cultured ginseng [2].

Nowadays, bioconversion, also called biotransformation, has been widely applied to plants containing bio-functional chemicals. Bioconversion through fermentation is a well-established traditional technology across history used to enhance nutritional properties. The presence of complex enzymes in lactic acid bacteria (LAB), such as *Lactobacillus* pp., *Pediococcus* spp., and *Leuconostoc* spp., leads to functional benefits, for instance, improving nutritional value, digestibility of foods, and modification of flavor attributes [3,4] 

According to a previous study [5], *Leuconostoc mesenteroides* KCCM 12010P isolated from kimchi has a high level of β-glucosidase activity, which can convert ginsenoside Rb2 to Rd and Rg3. In that study, ginseng extract showed anti-inflammatory and anticancer effects following fermentation by *L. mesenteroides*. Polyphenol content also can be bioconverted by *Lactobacillus plantarum*, as in the study of Gan et al. [6], where 9 h of fermentation by *L. plantarum* WCFS1 increased total polyphenol content and improved the antioxidant capacity in mung bean.

Oxidative stress, caused by free radicals, gives rise to peroxidation of proteins, lipids, and DNA, and ultimately induces inflammatory responses, resulting in chronic diseases [7,8]. Obesity, a major public health problem, is also associated with an inflammatory disorder [9]. Adipocytes do not only store surplus energy, but also play a crucial role in the endocrine and immune systems that regulate human body homeostasis. For example, mature adipocytes can up-regulate the transcriptional regulator, nuclear factor kappa-light-chain-enhancer (NF-κB), leading to the secretion of interleukin-6 (IL-6) and the tumor necrosis factor-α (TNF-α), which moderate immune response [10,11]. Several studies have reported that TNF-α plays a major role in insulin resistance and is associated with obesity, type 2 diabetes and adipocyte differentiation [12]. Previous studies have confirmed the biofunctionality of fermented ginseng, such as its anti-hyperglycemic effect on type 2 diabetes mellitus mice or its inhibitory effect on inducible nitric oxide synthase (iNOS) expression in RAW 264.7 cells [13,14]. However, we do not know of any studies on hydroponic ginseng, including shoots and roots fermented with lactic acid bacteria.

In this study, each part of hydroponic ginseng was fermented with *Leuconostoc mesenteroides* KCCM 12010P isolated from kimchi, and the resultant enhanced total polyphenol content and health-improving properties, including antioxidant, anti-inflammatory, and anti-adipogenic effects, were verified.

## 2. Results and Discussion

### 2.1. Enzymatic Activity of Leuconostoc Mesenteroides KCCM 12010P

The enzymatic activity of the strain was measured with the API ZYM kit (Biomerieux Inc., Marcy I’Etoile, France), and indicated as 0, 5, 10, 20, or 40 nM as indicated in a manual of the kit. The strain had ˃40 nM β-glucosidase activity, 5 nM esterase, alkaline phosphatase, and acid phosphatase activity, 10 nM leucine arylamidase, Naphthol-ASBI-phosphohydrolase, 20 nM α-galactosidase, and 30 nM α-glucosidase activity in hydrolyzed substrate. However, β-glucuronidase, which is a factor in developing cancer was not detected (Table 1).

### 2.2. Changes in the Number of Cells and pH During Fermentation

The viable cell number in HPG roots fermented by *L. mesenteroides* KCCM 12010P increased from 6.14 to 7.50 Log CFU/ mL during 24 h fermentation, while the microbial growth in fermented HPG shoots increased from 6 to 7.6 Log CFU/ mL (Figure 1). The pH in broth containing 5% HPG root and 5% HPG shoot extracts decreased from 5.95 to 4.73 and 5.82 to 4.39, respectively, which could be attributed to the bioconversion of phytochemicals, such as polyphenols and ginsenosides in the samples [15].

### 2.3. Total Polyphenol and Flavonoid Content

As shown in Table 2, the total polyphenol content (TPC) and total flavonoid content (TFC) of HPG roots and shoots increased during fermentation. The TPC of fermented HPG roots and shoots increased by 107.19% and 123.19%, respectively, after fermentation. Flavonoid (a kind of polyphenolic compound) contents were also increased by 645.59% and 217.98%, in roots and shoots, respectively. According to a previous study by Rodríguez et al. [16], these results could be due to enzymatic degradation of rutin (quercetin-3-*O*-rutinoside), which is abundant in ginseng, to a kaempferol-3-rutinoside intermediate during fermentation.

### 2.4. Antioxidant Activity of Fermented Hydroponic Ginseng

The 2,2′-azino-bis-3-ethylbenzothiazoline-6-sulfonic acid (ABTS), β-carotene, and ferric reducing antioxidant power (FRAP) assays were performed in this study to determine antioxidant activity. The ABTS cation accepts a hydrogen radical or electron to change to a stable diamagnetic molecule and reduced free radical. Table 2 shows the ABTS radical scavenging activities of roots and shoots extracts in fermented and non-fermented HPG. The ABTS radical scavenging activity increased from 34.21% to 46.41% and 54.69% to 62.30% in fermented hydroponic ginseng roots and shoots, respectively. The reducing power of the HPG extracts was also increased by fermentation. The HPG FRAP assay shows an increase in the ferric ion reducing power from 89.47 to 110.09 μg Fe^2+^/mg sample and from 214.76 to 222.06 μg Fe^2+^/mg sample in HPG roots and shoots, respectively (Table 2). β-Carotene-linoleic acid assay showed the lipid oxidation inhibitory activity of samples. Inhibition of linoleic acid oxidation of HPG roots and shoots increased slightly (by 102.7–104.1%) after fermentation. The different results of three methods may be due to different principles of antioxidant assay methods and, in addition, different contents of ginsenosides as well as polyphenols such as chlorogenic acid, gentisic acid, and coumaric acid according to part of ginseng plant [17]. Overall, the antioxidant activities of HPG shoots were greater than those of roots, and the antioxidant activities increased even after fermentation. Lee et al. [18] reported that the radical scavenging activities of ginseng leaves and stems were not affected by fermentation, but there were significant increases in phenolic compounds, which is contrary to our observations in this study. As TPC and TFC increased during fermentation, the antioxidant activity of fermented samples increased. It is considered that these results may occur due to the increase in low molecular weight polyphenol compounds and enzymatically bioconverted compounds produced by fermentation of *L. mesenteroides* KCCM 12010P.

### 2.5. Anti-Inflammatory Effect of Non-fermented and Fermented Hydroponic Ginseng

In this study, the amount of nitric oxide (NO) produced from the RAW 264.7 cells, as a form of NO^2-^ present in the cell medium, was measured and RT-PCR was performed to measure the expression of iNOS and cytokines, including IL-6, IL-1β, and TNF-α. It has been reported that roots and shoots of hydroponic ginseng, as well as ginseng, exhibit the inhibitory effects of inflammation in lipopolysaccharide (LPS)-stimulated RAW 264.7 cells [2]. To evaluate the anti-inflammatory activity of non-fermented and fermented HPG, cytotoxicity in RAW 264.7 cells was preliminarily assessed using a 3-[4,5-dimethylthiazol-2-yl]-2,5 diphenyl tetrazolium (MTT) assay. No cytotoxicity was observed in samples at treatment concentrations, i.e., 100–200 μg/mL (data not shown), but there was a significant decrease in NO production after HPG fermentation (Figure 2). NO concentration in roots and shoots decreased by 54.83% and 35.73% at 200 μg/mL of fermented HPG, respectively, compared to non-fermented samples. LPS in the extracellular membrane of gram-negative bacteria has been known to increase proinflammatory cytokines, such as TNF-α, IL-6, and interleukin-1β (IL-1β), which are transcribed by NF-κB [19]. In a previous study [5], *L. mesenteroides* KCCM 12010P degraded ginsenoside Rb1 to Rd, compound K, and Rh2 in a stepwise manner. It seems that the amount of Rd, compound K and Rh2 in samples may increase due to bioconversion since HPG is rich in Rb1 and Rd. According to the study of Cho et al. [20], ginsenoside Rb1 inhibits TNF-α expression by suppression of NF-κB activation. The increased ginsenoside Rd, which has a suppressive effect on inflammatory responses in later stages, after ischemia, by inhibiting iNOS and COX-2 [21,22], is considered as a factor that helps to reduce NO generation (Figure 3a and Figure 4a). After fermentation, TNF-α and IL-1β expression was suppressed in the root extract treated group. In the shoot extract treated group, cytokine expression was not significantly decreased, compared with levels prior to fermentation, but the inhibition rate remained at a low level (Figure 4). As shown Figure 4 and Figure 5, it appeared that a treatment with fermented ginseng root increased anti-inflammatory effects while, in case of shoots, fermentation decreased the anti-inflammatory effect. In the HPG plant, there are many kinds of bio-active phytochemicals and each composition may be different in between root and shoot. During fermentation, some compounds can be bioconversed or the contents of each compound can be changed [5.6]. Therefore, although the compounds have the anti-inflammatory effects, the degree can be different. The major compounds for anti-inflammation in fermented HPG are still unknown. 

From the macroscopic point of view, it seems that the prevention of diabetes, cancer, and cardiovascular diseases caused by chronic inflammation may be possible through the inhibitory mechanisms against such inflammation in fermented HPG.

### 2.6. Anti-Adipogenic Activity of Non-fermented and Fermented Hydroponic Ginseng

The effect of non-fermented and fermented HPG on lipid accumulation during adipogenesis in 3T3-L1 cells was measured by Oil Red O (ORO) staining. Each group was treated with 100–200 μg/mL of non-fermented and fermented HPG root, shoot extract. As described in previous studies, lipid droplets appeared on the second day of differentiation and increased until day nine (data not shown) [23]. ORO staining showed that the differentiation induced by MID, a mixture of insulin + dexamethasone + 3-isobutyl-1-methylxanthine (IBMX), occurred successfully (Figure 5). In the case of a group treated with fermented root, it appeared that fermentation with *L. mesenteroides* KCCM 12010P significantly decreased lipid accumulation during the nine-day incubation. Meanwhile, the cell viability of samples was measured via the MTT assay and it was shown that the samples (100–200 μg/mL) did not affect the viability of 3T3-L1 cells (data not shown). From these data, it appeared that lipid accumulation was inhibited by approximately 66% of the differentiated group when treated with 200 μg/mL of root extract before fermentation (Figure 5). Chae et al. [8] reported that several cytokines, such as TNF-α, IL-6, and NF-κB, involved in immune response are also secreted by adipocytes and regulate food intake, energy expenditure, and hormone functions. Therefore, the inhibition of lipid accumulation by fermented HPG roots is closely related to the inhibitory effects of these cytokines (Figure 3 and Figure 4). Hence, HPG shoots with high inflammatory cytokine inhibition rates are expected to have higher inhibitory effects on lipid accumulation in 3T3-L1 cells, but the results obtained here contradict this hypothesis. Although 200 μg/mL fermented shoot extract significantly decreased lipid accumulation by 76.08% of the differentiated control group, it has a lower inhibition rate compared to the root extract-treated group. The carbohydrate content in root and leaves are 59% and 73%, respectively [24,25], and a high sugar concentration is considered to affect lipid accumulation. In one study [26], the supplementation of a high-fat diet with hydroponic ginseng vinegar, containing ginsenoside Rg2, resulted in decreased body weight, fat weight, and liver weight in mice.

## 3. Materials and Methods 

### 3.1. Materials

Two-year-old whole hydroponic ginseng (HPG) were purchased from Chungjung–Saessacksam Company, Gwangju, Republic of Korea. RAW 264.7 cells were purchased from Korean Cell Line Bank (Seoul, Korea) and 3T3-L1 cells were taken from American Type Culture Collection (ATCC), Rockville, USA. The media, Man Rogosa Sharpe (MRS) broth was purchased from Difco Laboratories, Detroit, MI, USA. The reagents, 3-[4,5-dimethylthiazol-2-yl]-2,5 diphenyl tetrazolium (MTT), insulin, dexamethasone, lipopolysaccharide (LPS), 3-isobutyl-1-methylxanthine (IBMX), and Oil Red O (ORO) were purchased from Sigma-Aldrich Co., St. Louis, MO, USA. Dulbecco’s modified Eagle’s medium (DMEM), penicillin-streptomycin solution, and fetal bovine serum (FBS) were purchased from Hyclone^TM^ (Logan, UT, USA).

### 3.2. Microorganisms

*L. mesenteroides* KCCM 12010P was screened and isolated from white cabbage kimchi and identified by the Korean Culture Center of Microorganisms (KCCM, Seoul, Korea). The strain was cultured in MRS broth at 37 °C for 24 h before inoculation into HPG samples.

### 3.3. Determination of Bacterial Production of Enzymes 

Enzyme produced by *L. mesenteroides* KCCM 12010P was determined using API ZYM kit (Biomerieux Inc., Marcy l’Etoile, France) according to the company’s manual. A 75 μL of 10^7^ CFU/mL of the strain in phosphate buffered saline (PSB) was inoculated into each cupule at 37 °C for 4 h. A sample of empty cupule was used as a control. ZYM-A and ZYM-B reagents were then mixed and a color change of the solutions was observed. Enzyme activities during culturing were expressed as 0, 5, 10, 20, or 40 nM, as indicated in the API ZYM kit. 

### 3.4. Preparation for Hydroponic Ginseng Fermentation

Two-year-old HPG was separated into roots and shoots. Each part was lyophilized and extracted twice with 70% methanol in a reflux system. The extracts were concentrated using a vacuum evaporator (EYELA N-1000V, Tokyo, Japan) and then freeze-dried. pH of lyophilized extracts in distilled water at 5% (*w/v*) was adjusted to 6.5 to induce an optimum growth rate. The samples were autoclaved at 121 °C for 15 min and *L. mesenteroides* KCCM 12010P was inoculated into 5% broth of hydroponic ginseng roots and shoots. The initial number of cells was adjusted to approximately 6 Log CFU/mL. The non-fermented and fermented extracts were filtered with a 0.45 μm membrane filter and then used as samples during subsequent experiments.

### 3.5. Changes in Viable Number of Cells and pH during Fermentation

During fermentation, samples were taken from the cultures at 0, 3, 6, 9, 12, and 24 h to estimate the change in viable numbers of cells and pH. The bacterial count for *L. mesenteroides* KCCM 12010P on MRS agar was estimated using the plate count method after incubation at 37 °C for 1 d. The changes in pH were determined using a pH meter (WTW-720, Weilheim, Germany).

### 3.6. Measurement of Total Flavonoid Content

The total flavonoid content (TFC) was analyzed using the aluminum chloride assay described by Eom et al. [15]. The samples concentration was 5.0 mg/mL. TFC was calculated in milligrams of kaempferol equivalents (KPE) per g solid sample. The kaempferol calibration curve equation was as follows: Y = 0.0068X + 0.0426, R^2^ = 1(1)
where X is the amount of kaempferol content equivalent to gallic acid, Y is the absorbance at 415 nm.

### 3.7. Measurement of Total Polyphenol Content

The total polyphenol content (TPC) was measured using a slightly modified Folin–Ciocalteu method of Eom et al. [15]. The hydroponic ginseng shoot and root extracts were prepared in 5.0 mg/mL aliquots. Gallic acid was used as a standard and TPC concentration was expressed as milligrams of gallic acid equivalents (GAE) per g solid sample. The gallic acid calibration curve equation was as follows:Y = 0.0019X-0.0114, R^2^=0.9979(2)
where X is the amount of total flavonoid content equivalent to kaempferol, Y is the absorbance at 750 nm.

### 3.8. 2,2′-azino-bis-3-ethylbenzothiazoline-6-sulfonic acid (ABTS) Radical Scavenging Assay

An ABTS radical scavenging assay was performed as described by Chung et al. [17], with subtle modifications. A radical cation solution of ABTS was prepared with a 1:1 mixture of 14 mM ABTS and 5 mM potassium persulfate (in 0.1 M potassium phosphate buffer pH 7.4). The mixture was incubated at 25 °C for 16 h. The mixture was then diluted with 0.1 M potassium phosphate buffer till the absorbance was 0.7 (0.7 ± 0.02) at a wavelength of 734 nm. The antioxidant rate was calculated as follows:(3)$$\mathrm{Antioxidant}\;\mathrm{rate}\;(\%)\;=\;\left(1-\frac{A_{s}}{A_{c}}\right)\times 100$$
where A_c_ and A_s_ are the absorbance for a control (distilled water) and sample, respectively.

### 3.9. Ferric Reducing Antioxidant Power (FRAP) Assay

The ferric ion reducing effect of samples were determined via a modified FRAP assay method [27]. The cocktail solution was made by mixing acetate buffer (300 mM, pH 3.6), 10 mM 2,4,6-tris (2′-pyridyl)-s-triazine in 40 mM HCl, and 20 mM ferric chloride hexahydrate solution at a ratio of 10:1:1 in a 37 °C water bath. The sample (50 μL) was added to 950 μL cocktail solution and incubated under a dark condition at 25 °C for 30 min. Fe^3+^ to Fe^2+^ reduction was monitored by measuring the absorbance at 593 nm. Ferrous sulfate heptahydrate (FeSO_4_ ·7H_2_O) was used as a standard reagent.

### 3.10. β-Carotene-linoleic Acid Assay

A β-carotene-linoleic acid assay was used to determine the inhibitory effect on lipid oxidation of samples. The procedure was carried out using a slightly modified method of Eom et al. [5]. β-carotene (2 mg), linoleic acid (44 μL), and Tween 80 (200 μL) were mixed thoroughly in 10 mL chloroform. The mixture was evaporated with a rotary evaporator (EYELA N-1000V, Tokyo, Japan) at 42 °C. The residue was dissolved in 100 mL of distilled water and mixed to form an emulsion. The absorbance of this sample was controlled to 1.8–2.0 at 470 nm. A 500 μL sample was added to 4.5 mL aliquots of the emulsion. After reaction at 50 °C for 3 h, the absorbance was measured at 470 nm. The inhibition rate of β-carotene-linoleic acid oxidation was determined using the following equation:(4)$$\mathrm{Inhibition}\;\mathrm{rate}\;\mathrm{of}\;\beta \text{-}\mathrm{carotene}\text{-}\mathrm{linoleic}\;\mathrm{acid}\;\mathrm{oxidation}\;(\%)\;=\;\frac{A_{t}}{A_{0}}\times 100$$
where A_0_ and A_t_ is the absorbance value at the start of incubation for sample and the absorbance after 3 h of reaction, respectively.

### 3.11. Measurement of Nitric Oxide (NO) Production in RAW 264.7 Macrophage Cells

Analysis of NO produced in RAW 264.7 cell was carried out using the method of Cha et al. [28]. The RAW 264.7 cells were seeded in a microplate. The cell numbers were adjusted to 2 × 10^5^ cells/well. After incubation for 2 h at 37 °C, the cells were treated with various concentrations of samples and stimulated with 0.1 μg/mL of LPS for 24 h. Each medium was transferred to another microplate and the nitrite concentration was determined by adding 100 μL of Greiss reagent. The absorbencies were detected at 570 nm with an ELISA reader (Emax, Torrance, CA, USA). The amount of nitrite present was calculated from a sodium nitrite (NaNO_2_) standard curve.

### 3.12. Measurement of Expression of Inflammation-Related Mediators

Total mRNA from RAW 264.7 cells was extracted using a RNeasy^®^ Mini Kit, QIAGEN, Hilden, Germany. The cDNA was synthesized by a RevertAid First Strand cDNA Synthesis Kit, Thermo Fisher Scientific, UK, followed by a reverse transcription-polymerase chain reaction (RT-PCR), using a SensiFAST™ SYBR No-ROX Kit (Bioline, London, UK). The primers used in this study are presented in Table 3.

### 3.13. Cell Viability Assay

Cell viability was assayed through MTT assay using the 3T3-L1 preadipocyte [29]. The cells were inoculated in a 96-well microplate at a concentration of 1 × 10^5^ cells per well, and incubated for 1 d at 37 °C. The cells were treated with 100 μL fermented HPG samples. After 44 h of incubation, the media in the wells were discarded. MTT solution (50 μL) was added and incubated for 4 h, after which 50 μL dimethyl sulfoxide (DMSO) was also added per well to dissolve the crystals. Absorbance was measured at 570 nm using an ELISA reader (Emax, Torrance, CA, USA).

### 3.14. Cell Culture and Pre-Adipocyte Differentiation

The 3T3-L1 cell line was maintained and differentiated using modified methods of Nam et al. [30]. They were seeded in a 12-well plate (2 × 10^5^ cells/well) in DMEM with 1% penicillin-streptomycin solution and 10% FBS. Prior adipocyte differentiation induction, 3T3-L1 cells were grown to post-confluence, and differentiation was induced by DMEM mixture, containing 10% FBS, 115 μg/mL IBMX in 0.35 M KOH, 10 μM dexamethasone in ethanol (day 0), 10 μg/mL insulin in 0.1 N filtered HCl, and different sample concentrations. After 3 days (day 3), the culture medium was replaced with other DMEM mixture, containing 10% FBS and 10 μg/mL insulin. After another 3 days (day 6), the medium was replaced with DMEM mixture, only containing 10% FBS. By day 9, the cells were fully differentiated and were used for Oil Red O staining.

### 3.15. Oil Red O Staining

On the ninth day of differentiation above, accumulated lipids in adipocyte cells were stained using the modified method described by Nam et al. [30] to detect the rate of adipogenesis. Cells were washed mildly using phosphate buffered saline (PBS) and fixed with 1 mL 10% formalin for 1 h in the dark. After washing with formalin, fixed cells were thoroughly dried in a hood. One milliliter of ORO working solution (ORO stock solution: Water = 6:4) was treated for 20 min. Then, cells were washed using distilled water and observed with an Olympus IX51 inverted microscope with Camedia C-4040 camera (Center Vally, PA, USA). Isopropanol (500 μL) was added to elute stained lipids from cells and the absorbance was measured at 504 nm.

### 3.16. Statistical Analysis

All experiments were repeated three times. The values were presented as mean ± standard deviation (SD). Using these data, one-way analysis of variance (ANOVA) was performed using SPSS software (version 20, SPSS Inc., Chicago, IL, USA). A Duncan’s multiple range test was used to test for significant differences between treatments at *p* < 0.05. 

## 4. Conclusions

This study aimed to evaluate the antioxidant, anti-inflammatory, and anti-adipogenic activities of hydroponic ginseng root and shoot fermented by lactic acid bacteria. During 24 h fermentation in 5% broth of HPG-root and HPG-shoot extract, *L. mesenteroides* KCCM 12010P growth was not affected significantly. Furthermore, HPG roots and shoots fermented by *L. mesenteroides* KCCM 12010P showed significant increments in total polyphenolic compounds and flavonoid contents due to bioconversion driven by enzymes, including β-glucosidase. Antioxidant activity was enhanced in both fermented samples, resulting in increased phenolic compound content. Fermented HPG suppressed the production of both iNOS and NO and the expression of inflammation-related cytokines, including TNF-α, IL-1β, and IL-6. Additionally, lipid accumulation was suppressed in 3T3-L1 cells when the fermented samples were treated. Meanwhile, Hsu et al. [31] reported that phenolic acids and flavonoids reduced intracellular triglyceride content by elucidating GPDH activity, which is involved in adipogenesis. From these results, *L. mesenteroides* KCCM 12010P improved HPG biofunctional activities, such as antioxidant, anti-inflammatory, and antiadipogenic effects, indicating that fermented HPG could be applied in the functional food or medicinal industry as a potent bioactive material for human health.

## Figures and Tables

**Figure 1 molecules-24-03359-f001:**
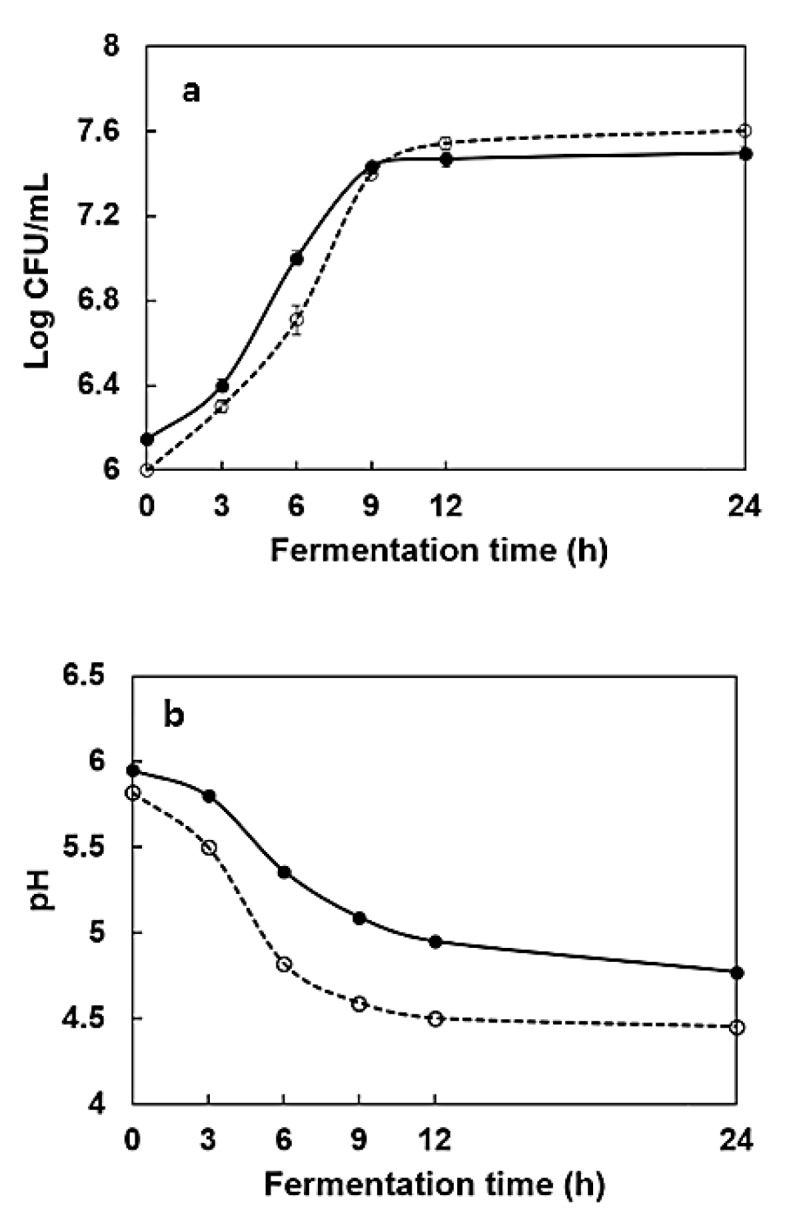
Growth curve (**a**) and pH (**b**) changes in hydroponic ginseng root and shoot during 24 h fermentation by *Leuconostoc mesenteroides* KCCM 12010P. Hydroponic ginseng (HPG) roots (●); HPG shoots (○).

**Figure 2 molecules-24-03359-f002:**
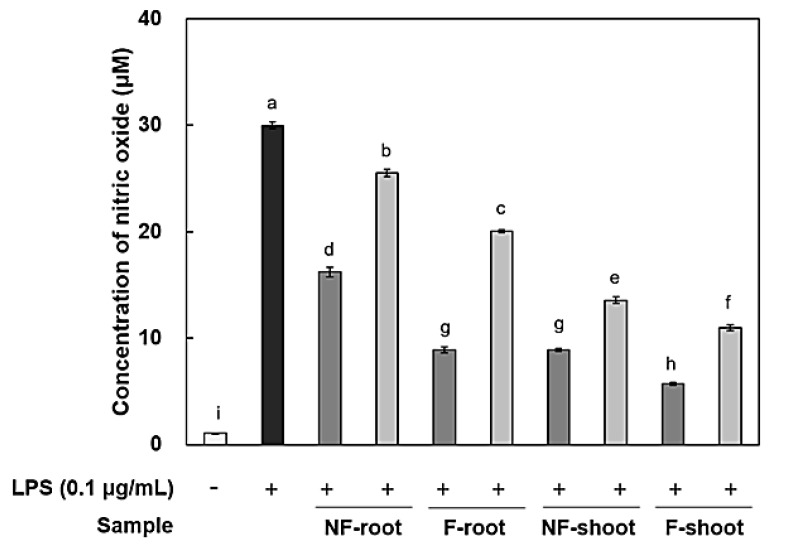
Anti-inflammatory effect in RAW 264.7 cells treated with hydroponic ginseng fermented with *Leuconostoc mesenteroides* KCCM 12010P. 200 μg/mL; 100 μg/mL. Values are means ± standard deviation of triplicate experiments. Different letters indicate significant differences as determined by Duncan’s multiple range tests (*p* < 0.05).

**Figure 3 molecules-24-03359-f003:**
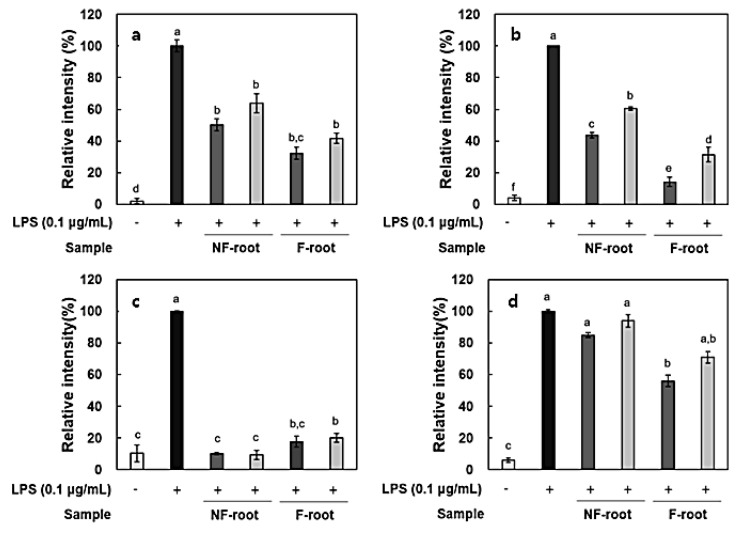
Relative expression of inflammatory mediators in lipopolysaccharide (LPS)-stimulated RAW 264.7 cells treated with hydroponic ginseng root. (**a**), Inducible nitric oxide synthase (iNOS); (**b**), tumor necrosis factor-α (TNF-α); (**c**), interleukin-6 (IL-6); and (**d**), interleukin-1β (IL-1β). 200 μg/mL; 100 μg/mL. NF, non-fermented; F, fermented. Values are means ± standard deviation of triplicate experiments. Different letters in same column indicate significant differences as determined by Duncan’s multiple range tests (*p* < 0.05). Different letters (a, b, c, and d) represent significant differences among samples.

**Figure 4 molecules-24-03359-f004:**
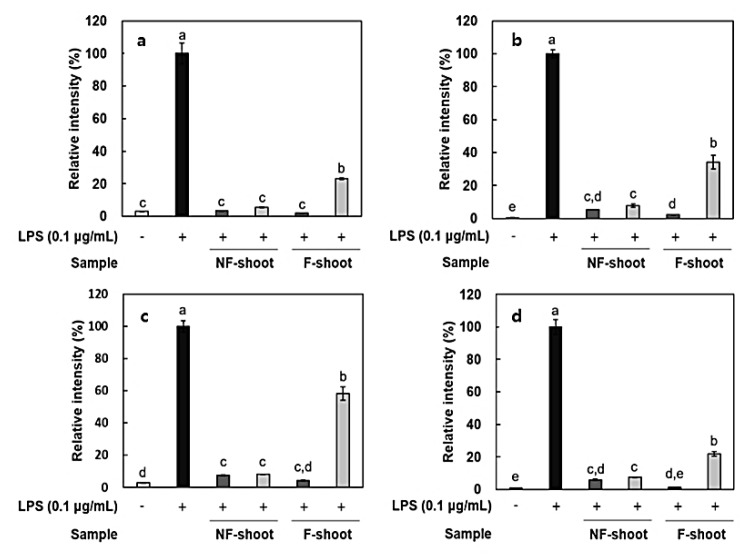
Relative expression of inflammatory mediators in LPS-stimulated RAW 264.7 cells treated with hydroponic ginseng shoot. (**a**), iNOS; (**b**), TNF-α; (**c**), IL-6; and (**d**), IL-1β. 200 μg/mL; 100 μg/mL. NF, non-fermented; F, fermented. Values are means ± standard deviation of triplicate experiments. Different letters in same column indicate significant differences as determined by Duncan’s multiple range tests (*p* < 0.05). Different letters (a, b, c, and d) represent significant differences among samples.

**Figure 5 molecules-24-03359-f005:**
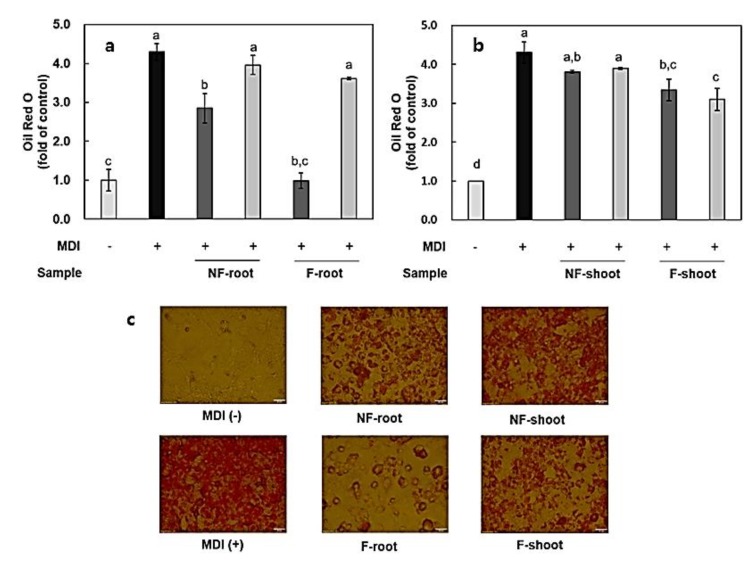
Effects of fermented hydroponic ginseng root (**a**) and shoot (**b**) on lipid accumulation in differentiated 3T3-L1 cells, and Oil Red O (ORO) stained cells after differentiation (**c**). 200 μg/mL; 100 μg/mL. NF, non-fermented; F, fermented; MDI, insulin + dexamethasone + 3-isobutyl-1-methylxanthine (IBMX). Values are means ± standard deviation of triplicate experiments. Different letters indicate significant differences as determined by Duncan’s multiple range tests (*p* < 0.05). Different letters (a, b, c, and d) represent significant differences among samples.

**Table 1 molecules-24-03359-t001:** Enzymatic activity of *Leuconostoc mesenteroides* KCCM 12010P using an API kit.

Enzymes	Activity ^1)^	Enzymes	Activity
Control	0	Acid phosphatase	5
Alkaline phosphatase	5	Naphthol-ASBI-phosphohydrolase	10
Esterase	5	α-Galactosidase	20
Esterase lipase	0	β-Galactosidase	≥40
Lipase	0	β-Glucuronidase	0
Leucine arylamidase	10	α-Glucosidase	30
Valine arylamidase	0	β-Glucosidase	≥40
Cystine arylamidase	0	*N*-Acetyl-β-glucosaminidase	0
Trypsin	0	α-Mannosidase	0
α-Chymotrypsin	0	α-Fucosidase	0

^1)^ 0, no enzyme activity; 5, 10, 20, 30, and ≥ 40 indicates concentration (nM) of hydrolyzed substrate after 4 h of incubation at 37 °C.

**Table 2 molecules-24-03359-t002:** Total polyphenol, flavonoid contents, and antioxidant activities of non-fermented and fermented hydroponic ginseng.

Sample		Total Polyphenol Content (GAE^1)^)	Total Flavonoid Content (KPE^2)^)	Antioxidant Activity		
				ABTS Radical Scavenging Activity (%)	FRAP Assay (μM Fe^2+^/mg Solid Sample)	Inhibition of β-carotene Oxidation (%)
Non- fermented	Roots	27.96 ± 0.35^d^	0.68 ± 0.18^d^	34.21 ± 0.36^d^	89.47 ± 0.865^d^	84.72 ± 0.72^b^
	Shoots	32.56 ± 0.40^b^	13.74 ± 0.24^b^	54.69 ± 1.20^b^	214.76 ± 0.28^b^	74.76 ± 0.67^c^
Fermented	Roots	29.97 ± 0.57^c^	4.39 ± 0.10^c^	46.41 ± 1.38^c^	110.09 ± 1.27^c^	88.20 ± 0.47^a^
	Shoots	40.11 ± 0.98^a^	29.95 ± 0.46^a^	62.30 ± 1.20^a^	222.06 ± 0.22^a^	76.76 ± 0.41^d^

All values were expressed as means ± standard deviation of triplicated experiments. Different letters in same column indicate significant differences as determined by Duncan’s multiple range tests (*p* < 0.05). ^1)^ GAE: mg gallic acid /g solid content of sample. ^2)^ KPE: mg kaempferol /g solid content of sample. ABTS: 2,2′-azino-bis-3-ethylbenzothiazoline-6-sulfonic acid, FRAP: Ferric reducing antioxidant power. Different letters (a, b, c, and d) represent significant differences among samples.

**Table 3 molecules-24-03359-t003:** Primer sequence of genes used for a reverse transcription-polymerase chain reaction.

	Forward	Reverse
β-actin	5′-GTCGGCCTAGGCACCAG-3′	5′-GGAGGAAGAGGATGCGGCAGT-3′
iNOS	5′-CCCTTCCGAACTTTCTGGCAGCAGC-3′	5′-GGCTGTCAGAGTCTCGTGGCTTTGG-3′
TNF-α	5′-GCAGAAGAGGCACTCCCCCA-3′	5′-GATCCATGCCGTTGGCCAGG-3′
IL-1β	5′-CAGGATGAGGACATGAGCACC-3′	5′-CTCTGCAGACTCAAACTCCAC-3′
IL-6	5′-AGTTGCCTTCTTGGGACTGA-3′	5′-TTCTGCAAGTGCATCGT-3′

iNOS, inducible nitric oxide synthase; TNF-α, tumor necrosis factor-alpha; IL-1β, interleukin-1beta; IL-6, interleukin-6.

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
