# Peer review of "Improved Antioxidant, Anti-inflammatory, and Anti-adipogenic Properties of Hydroponic Ginseng Fermented by *Leuconostoc mesenteroides* KCCM 12010P"

_molecules, 2019, doi:10.3390/molecules24183359_

Round 1
Reviewer 1 Report
This manuscript demonstrated that hydroponic ginseng fermented by a specific Leuconostoc mesenteroides KCCM 12010P strain (especially the fermented root) had better antioxidant, anti-inflammatory and anti-adipogenic properties compared to non-fermented hydroponic ginseng, Generally, the data were clearly presented. However, there are several confusion and conflicts in the ms. The major concerns related to the ms were listed below.
Line 42. In the introduction, the authors described that “However, very few studies, comparing the functional ……… to those of ginsengs grown in soil, exist.” It is not clear the meaning of this sentence. Particularly, there is no discussion or comparison between hydroponic ginseng and ginseng from soil in the experiment.
There is no description related to the enzymatic activity test although the authors did mention the APIZYM kit. The ms. should include the detail condition for cell culture condition, cell density and how to concentrate the extracellular protein and the levels of total proteins for the enzymatic analysis since all these factors may affect the results. Besides, there is no explanation of “control” in Table 1.
In Fig 1. The authors conclude that the changes of cell number and pH during fermentation were possibly attributed to the bioconversion of phytochemicals. In order to make this conclusion, the authors should also include a control experiment- cells were cultured exactly the same as experimental groups except without ginseng extract.
The main idea and conclusion of this ms. is the enhanced total polyphenol content and health-improving properties of fermented ginseng. However, in Line 132-138, the authors implied that the anti-inflammatory effects might be due to ginsenosides according to their previous published work (Eom et al., 2018). The authors should clearly explain the discrepancy between the effects of polyphenols and the ginsenosides.
Line 238: the formula (…. Where X is the amount of kaempferol) should be corrected to “the amount of total flavonoid content equivalent to kaempferol)
Author Response
Thanks for your kind reviewing. I revised some parts as you comment as possible and the revision in manuscript was indicated in red.
Responses on your comments
Line 42. In the introduction, the authors described that “However, very few studies, comparing the functional ……… to those of ginsengs grown in soil, exist.” It is not clear the meaning of this sentence. Particularly, there is no discussion or comparison between hydroponic ginseng and ginseng from soil in the experiment.
→ I revised this part (line 41-45).
There is no description related to the enzymatic activity test although the authors did mention the APIZYM kit. The ms. should include the detail condition for cell culture condition, cell density and how to concentrate the extracellular protein and the levels of total proteins for the enzymatic analysis since all these factors may affect the results. Besides, there is no explanation of “control” in Table 1.
→ I explained the API ZYM kit method in line 233-239. A cupule contains different reagent commercially according to a kind of enzyme. There is not any reagent in a cupule for control (empty cupule).
In Fig 1. The authors conclude that the changes of cell number and pH during fermentation were possibly attributed to the bioconversion of phytochemicals. In order to make this conclusion, the authors should also include a control experiment- cells were cultured exactly the same as experimental groups except without ginseng extract.
→ In our previous study, it appeared that there was no difference significantly in microbial growth effects of between sample treated with root- or shoot- HPG and no-treated sample (control).
The main idea and conclusion of this ms. is the enhanced total polyphenol content and health-improving properties of fermented ginseng. However, in Line 132-138, the authors implied that the anti-inflammatory effects might be due to ginsenosides according to their previous published work (Eom et al., 2018). The authors should clearly explain the discrepancy between the effects of polyphenols and the ginsenosides.
→ Polyphenol is one of bio-functional active component, but the content of polyphenols is very lower than that of ginsenosides in HPG for anti-inflammatory or immune-enforcing effect.
Line 238: the formula (…. Where X is the amount of kaempferol) should be corrected to “the amount of total flavonoid content equivalent to kaempferol)
→ I revised this part as you recommended (Line 267-268).
Reviewer 2 Report
Reviewer has read this manuscript with great interest. This work reports that hydroponic ginseng roots and shoots were fermented with Leuconostoc mesenteroides KCCM 12010P isolated from kimchi,. which the total phenolic compounds, antioxidant, anti-inflammatory, and anti-adipogenic effects were evaluated in comparison with non-fermented samples as a control group. The results show that fermented hydroponic ginseng has potential for applications in the functional food industry. The experimental and theoretical methods described comprehensively. The research contents are clearly reported and the conclusions are supported by the data in this manuscript. The abstract is well matched with the text contents. The manuscript is organized well. Overall, the reviewer thinks that the subject should be of interest to the readers of Molecules Journal, and this manuscript is suitable for publication in this journal in current form.
Author Response
Thanks for your kind reviewing our article.
Reviewer 3 Report
Ref: Molecules 575453
Title: Improved antioxidant, anti-inflammatory, and anti-adipogenic properties of hydroponic ginseng fermented by Leuconostoc mesenteroides KCCM 12010P
Generally, this study is appropriate designed and all the methods and results are clearly presented. It is acceptable for publication with minor revision.
Specific comments:
Line 129, please indicated which level for the decreasing by 54.83 and 35.73%. Line 138, Figure 4a or 4b? Compared to the antioxidant activities of roots, shoots exerted stronger ABTS radical scavenging activity and FRAP assay but inhibition of β-carotene oxidation (Table 2). Give the explanation for the observations. Line 154, check the legend. Cell viability and anti-inflammatory effect were not found in figure. Describe more about the reversed inflammatory effects of fermentation from root and shoot.
Author Response
Thank you for your kind reviewing our paper.
Our responses on your comments are:
Line 129, please indicated which level for the decreasing by 54.83 and 35.73%.
→ I added a concentration level (line 136).
Line 138, Figure 4a or 4b?
→ I corrected this part (line 146).
Compared to the antioxidant activities of roots, shoots exerted stronger ABTS radical scavenging activity and FRAP assay but inhibition of β-carotene oxidation (Table 2). Give the explanation for the observations.
→ I added the discussion in manuscript (line 116-119).
Line 154, check the legend. Cell viability and anti-inflammatory effect were not found in figure.
→ I corrected the legend in this figure (line 168).
Describe more about the reversed inflammatory effects of fermentation from root and shoot.
→ I discussed it in line 149-155.